# The non-selective Rho-kinase inhibitors Y-27632 and Y-33075 decrease contraction but increase migration in murine and human hepatic stellate cells

Nadine Bachtler[1], Sandra Torres[1], Cristina Ortiz[1], Robert Schierwagen[1], Olaf Tyc[1], Christoph Hieber[1], Marie-Luise Berres[2], Caroline Meier[1], Nico Kraus[1], Stefan Zeuzem[1], Bart Nijmeijer[3], Sebas Pronk[3], Jonel Trebicka[1,4]*, Sabine Klein[1]

1 Department of Internal Medicine I, Goethe University Frankfurt, Frankfurt, Germany, 2 Department of Medicine III, University Hospital RWTH Aachen, Aachen, Germany, 3 LinXis BV, Amsterdam, The Netherlands, 4 European Foundation for the Study of Chronic Liver Failure, Barcelona, Spain

* Jonel.trebicka@kgu.de

**Data Availability Statement:** All relevant data are within the paper and its Supporting Information files.

## Abstract

### Background

The Rho-kinase ROCK II plays a major role in the activation of hepatic stellate cells (HSC), which are the key profibrotic and contractile cells contributing to the development of chronic liver disease. Inhibition of ROCK II ultimately blocks the phosphorylation of the myosin light chain (MLC) and thus inhibits stress fibre assembly and cell contraction. We investigated the effects of the ROCK inhibitors Y-33075 as well as Y-27632 in murine and human hepatic stellate cells.

### Methods

Primary isolated HSC from FVB/NJ mice and the immortalized human HSC line TWNT-4 were culture-activated and incubated with Y-27632 and Y-33075 (10nM to 10μM) for 24h. Protein expression levels were analyzed by Western Blots and transcriptional levels of pro-fibrotic markers and proliferative markers were evaluated using real-time qPCR. Migration was investigated by wound-healing assay. Proliferation was assessed by BrdU assay. Contraction of HSC was measured using 3D collagen matrices after incubation with Y-27632 or Y-33075 in different doses.

### Results

Both Rho-kinase inhibitors, Y-27632 and Y-33075, reduced contraction, fibrogenesis and proliferation in activated primary mouse HSC (FVB/NJ) and human HSC line (TWNT-4) significantly. Y-33075 demonstrated a 10-times increased potency compared to Y-27632. Surprisingly, both inhibitors mediated a substantial and unexpected increase in migration of HSC in FVB/NJ.

**Funding:** This study was supported by the German Research Foundation (DFG) project ID 403224013 – SFB 1382 (A09), by the German Federal Ministry of Education and Research (BMBF) for the DEEP-HCC project and by the Hessian Ministry of Higher Education, Research and the Arts (HMWK) for the ENABLE cluster project and by Eurostars (Grant ID 12350). The MICROB-PREDICT (project ID 825694), DECISION (project ID 847949), GALAXY (project ID 668031), LIVERHOPE (project ID 731875) and IHMCSA (project ID 964590) projects have received funding from the European Union's Horizon 2020 research and innovation program. The manuscript reflects only the authors' views, and the European Commission is not responsible for any use that may be made of the information it contains. The funders had no influence on study design, data collection and analysis, decision to publish, or preparation of the manuscript.

**Competing interests:** The authors have declared that no competing interests exist.

**Abbreviations:** αSMA, α-smooth muscle actin; BDL, bile duct ligation; BrdU, Bromodesoxyuridine; Col1a1, collagen 1a1; ECL, enhanced chemiluminescence; ET-1, Endothelin-1; GAPDH, Glyceraldehyde 3-phosphate dehydrogenase; GEF, guanine exchange factor; GPCR, G-protein coupled receptor; HSC, hepatic stellate cel; KD, knock down; MLC, myosin light chain; MLCP, myosin light chain phosphatase; PCNA, proliferating cell nuclear antigen; PDGF, platelet-derived growth factor; PHT, portal hypertension; qPCR, quantitative polymerase chain reaction; ROCK, Rho-kinase; RT-qPCR, real-time quantitative polymerase chain reaction; SDS-PAGE, sodium dodecyl sulfate polyacrylamide gel electrophoresis; SEM, standard error of the mean; TBST, Tris-buffered saline with Tween20; TGFβ, transforming growth factor β.

## Conclusion

ROCK inhibition by the tested compounds decreased contraction but increased migration. Y-33075 proved more potent than Y27632 in the inhibition of contraction of HSCs and should be further evaluated in chronic liver disease.

## Introduction

Chronic liver disease is a major health problem affecting approximately 1.5 billion people worldwide and leading to 2 million deaths a year [1, 2]. Liver fibrosis is becoming more prevalent. It has various etiologies but invariably progresses towards a uniform end stage for which no treatment options exist [1, 3].

HSCs play a crucial role in the development of liver fibrosis. Due to (chronic) liver injury HSCs are activated and consequently transdifferentiate into a myofibroblast-like phenotype [4, 5]. In contrast to quiescent HSCs, these activated cells exhibit an increase in migration as well as proliferation, and produce large amounts of extracellular matrix proteins (collagen) and pro-fibrotic cytokines (e.g. transforming growth factor β, TGFβ; platelet-derived growth factor, PDGF) thus driving fibrosis [6, 7].

The accumulation of collagen in the liver, as well as the exaggerated contractile response of these myofibroblasts to vasoconstrictors cause an increased intrahepatic vascular resistance which results in the development of portal hypertension (PHT) in liver fibrosis [8].

The Rho-kinase is crucial in regulating cytoskeletal proteins and thus has a decisive effect on activation, contraction and fibrogenesis in HSCs [9, 10]. The Rho family of small G proteins comprises 20 members. RhoA, which is one of the small G proteins, is responsible for regulating the actin cytoskeleton and for mechanical forces [11]. Activated by guanine exchange factors (GEFs), RhoA activates Rho-kinase 2 (ROCK II) at the C-terminal coiled-coil domain [12]. The Rho-associated coiled-coil-forming kinase (Rho-kinase, ROCK) is one the most widely studied downstream effector kinases of the Rho-GTPase. Many substrates of ROCK are known, including the myosin light chain (MLC) and other myosin family proteins, among others [13].

In HSCs, MLC is of major importance for cellular contraction, stress fiber formation, focal adhesion and migration through ROCK.

The inhibition of ROCK has been shown to reduce both contraction and activation of HSCs by lowering phosphorylation of MLC and decreasing the expression of α-smooth muscle actin (αSMA), respectively [14–16]. ROCK inhibitors have also been shown to reduce fibrogenesis in HSCs [17]. Similarly, Y-27632 has demonstrated the ability to reduce portal pressure *in vivo* in cirrhotic rats thus making ROCK inhibitors a prime therapeutic target in the treatment of chronic liver disease [18, 19].

The present study examines the effects of the ROCK inhibitor Y-33075 compared to Y-27632 on activated HSCs, after their transdifferentiation with particular emphasis on contraction, proliferation and migration to provide clearer insights into the underlying mechanism of action.

## Materials and methods

### FVB/NJ mice

All experiments were performed in accordance with the German animal protection law and guidelines and were approved by the regional authority, the Hessian Animal Care and Use

Committee (FK/2005). Mice were purchased by Charles River Laboratories Research Model and Services Germany, Sulzfeld, Germany. Before liver perfusions, mice received ketamine/ xylazine anesthesia (100 mg ketamine/kg body wt and 10 mg xylazine/kg body wt) that was injected intraperitoneally.

### Isolation of murine hepatic stellate cells

HSCs from healthy male FVB/NJ mice were isolated as previously described [20]. In short, a two-step protocol consisting of sequential *in-situ* pronase E (Merck, Darmstadt, Germany) and collagenase D (Roche, Mannheim, Germany) perfusion of livers was performed followed by density gradient fractioning of the obtained cell suspensions using Nycodenz (AXIS-SHIELD, Norway). In order to ensure activation and transdifferentiation, primary HSC were seeded in uncoated plastic culture dishes [21] in Dulbecco's Modified Eagle Medium (DMEM (1x) (Gibco, ThermoFisher Scientific, Darmstadt, Germany)) supplemented with 20% fetal calf serum (FCS, Gibco, ThermoFisher Scientific, Darmstadt, Germany), 2% L-Glutamine (200nM) (Gibco, ThermoFisher Scientific, Darmstadt, Germany) and 1% Penicillin-Streptomycin (10000U/mL) (Gibco, ThermoFisher Scientific, Darmstadt, Germany), which was renewed every 48-72h. Cells were cultured for at least 7 days before experiments were performed. For the human derived TWNT-4 cell line, (kindly provided by Berres ML), DMEM 1x medium supplemented with 10% FCS, 2% L-Glutamine and 1% Penicillin-Streptomycin (10000U/mL) was used.

### Incubation of primary mouse HSCs, as well as human derived HSCs (TWNT-4) with the ROCK inhibitors Y-27632 and Y-33075

For these experiments Y-27632 dihydrochloride (Tocris, Wiesbaden, Germany) or Y-33075 (MedChemExpress, BIOZOL, Eching, Germany) was dissolved in PBS to create a stock solution at a concentration of 100mM. Cells were starved with medium containing DMEM 1x, 0% FCS, 2% L-Glutamine and 1% Penicillin-Streptomycin (10000U/mL) for 24 hours prior to incubation. The stock solution was further diluted in DMEM 1x medium augmented with 10% FCS for TWNT-4 or 20% FCS for FVB/NJ cells as well as 2% L-Glutamine and 1% Penicillin-Streptomycin to a final concentration of 10nM, 100nM, 1μM or 10μM and subsequently added to the starved cells for 24 hours. The same medium with PBS instead of the inhibitors was used to provide (no-inhibitor) controls.

### HSCs contraction assay

$2.5 \times 10^4$ primary HSCs or TWNT-4 cells were allowed to adhere to hydrated collagen gels to measure their contraction, as described previously [20]. The cells were treated with Y-27632, Y-33075 or PBS as a control. Pictures were taken at 0, 24, 48 and 72h, total gel area was calculated using ImageJ (version 1.51q, NIH, USA) and compared to the controls.

### Wound healing assay

Cell migration was assessed using a wound healing assay as described previously [22]. Briefly, cells were cultured in a 6-well plate until fully confluent. Cells were starved using a solution of DMEM 1x medium supplemented with 0% FCS, 2% L-Glutamine and 1% Penicillin-Streptomycin (10000U/mL) for 24 hours. Using a 10μL pipette tip a scratch was lined through the cell layer and pictures were taken at 0h, 4h, 8h and 24h with the microscope (AxioVert 200M, Zeiss, Heidelberg, Germany). The width of the scratch was measured with the ImageJ software (version 1.51q, NIH, USA) and compared to the initial picture.

### 5'-Bromo-2'-Deoxyuridine (BrdU) proliferation assay

Proliferation in HSCs was measured using the colorimetric BrdU ELISA kit (Roche, Mannheim, Germany) in accordance with the manufacturers' specifications. In short, cells were seeded at a density of $6 \times 10^5$ cells/ml in a 96-well plate and incubated with Y-27632 or Y-33075 for 24h. HSC were labelled with 10 μM BrdU for 2 hours and its incorporation was measured according to the guidelines.

### Quantitative Real-Time PCR (RT-PCR)

TRIzol-based RNA extraction [23] was performed on HSCs after 24 hours of incubation with Y-27632 or Y-33075. RNA concentration was measured and reverse transcription of total RNA was performed as previously described [24]. The TaqMan assays used were provided by Applied Biosystems (Foster City, USA). 18S rRNA served as endogenous control. Results were expressed as $2^{-\Delta\Delta CT}$, which corresponds to the x-fold increase of gene expression of the reference group. List of validated TaqMan® gene expression assays: *Col1a1 (*Hs00164004_m1), *Pcna (*Hs00427214_g1) and *Tgfβ (*Hs00998133-m1).

### α-SMA, phospho-Moesin and Col1a1 western blotting

Samples (30μg of protein/lane) were subjected to sodium dodecyl sulfate polyacrylamide gel electrophoresis (SDS-PAGE; 10% gel) and proteins were blotted on nitrocellulose membranes. Ponceau-S staining ensured equal protein loading. The membranes were blocked with 5% milk/TBST, incubated with primary antibodies and corresponding peroxidase coupled secondary antibodies (S1 Table) (Calbiochem, San Diego, CA). Glyceraldehyde-3-phosphate dehydrogenase (GAPDH) served as an endogenous loading control. Blots were developed with enhanced chemiluminescence (ECL, Amersham, UK). Intensities of the detected bands were quantified, and results were corrected for GAPDH levels using ImageJ (version 1.51q, NIH, USA).

### Phopho-MLC and MLC western blotting

TWNT-4 cells (approximately $2^*10^5$ cells per well) were seeded in a 12-wells tissue culture plate (Nunclon™ Delta Surface, Thermo Fisher Scientific, Bleiswijk, the Netherlands) and incubated with PBS or with different concentrations of Y-27632 or Y-33075. After 24 hours, cell lysates were prepared by resuspending the cells in 50 μl 1x Laemmli protein sample buffer with DTT. Next, the lysates were boiled for 10 minutes at 100°C, loaded on a 8–12% (w/v) PAGE gel (Bio-rad, Veenendaal, the Netherlands) and blotted onto a PVDF membrane (Roche, Mannheim, Germany). The blots were blocked with 1% BSA-TBS and thereafter stained overnight for phosphorylated MLC using a rabbit polyclonal antibody or for MLC using a rabbit polyclonal antibody followed by a 1 hour incubation with IRDye800-conjugated goat-anti-rabbit secondary antibody (LI-COR Biosciences, Lincoln, NE, USA). As a loading control, the blots were also stained for tubulin using a monoclonal mouse-anti-tubulin antibody followed by a 1 hour incubation with an IRDye680-conjugated goat-anti-mouse secondary antibody (LI-COR Biosciences, Lincoln, NE, USA). Fluorescent signal was detected using the Odyssey near-infrared scanner (LI-COR Biosciences, Lincoln, NE, USA).

### Statistical analysis

Statistical analyses of data were performed using Prism V.5.0 (GraphPad, San Diego, CA). Data were expressed as mean ± standard error of the mean (SEM). Comparisons between groups were done by non-parametric Mann–Whitney U t-tests. P<0.05 was considered as statistically significant.

## Results

### ROCK inhibitors reduce contraction in both human and murine HSCs

In order to reduce contraction in TWNT-4 cells as well as in primary FVB/NJ mouse HSCs, ROCK inhibitors were used. These ROCK inhibitors inhibit the contractile GPCR Pathway of RhoA and ROCK. Both ROCK inhibitors act on both ROCK I and II (Fig 1A). The molecular structure of the used ROCK inhibitors, Y-27632 and Y-33075, are analogous of one another (Fig 1B).

To analyze the effect of the ROCK inhibitors Y-27632 and Y-33075 on contraction, human TWNT-4 cells were plated to adhere to collagen gels before being treated with different concentrations of Y-33075 or Y-27632 (100nM, 1μM, 10μM) for 24h.

Contraction of TWNT-4 cells was significantly reduced with both inhibitors in a dose dependent manner. The ROCK inhibitor Y-33075 significantly inhibited contraction of TWNT-4 cells at concentrations ranging from 100nM to 10μM. Y-27632 inhibited contraction only at the higher concentrations of 1μM and 10μM. Thus, with respect to contraction, Y-33075 showed at least a 10-fold higher potency in TWNT-4 cells than Y-27632 (Fig 1C and 1D).

Primary mouse HSCs isolated from FVB/NJ mice were cultured in uncoated plastic culture dishes for 7 days to ensure activation and transdifferentiation before use in the contraction assay. Similarly to TWNT-4 cells, in FVB/NJ HSCs, Y-33075 decreased the contraction at concentrations of 1μM and 10μM. Further, we observed a tendency towards reduction at a concentration of 100nM. Y-27632 also significantly reduced contraction of FVB/NJ HSCs at a concentrations of 1μM and above (Fig 1E and 1F).

Moreover, we analyzed the phosphorylation levels of myosin light chain (p-MLC) as they are of major importance in ROCK mediated cellular contraction in HSCs. At concentrations as low as 10nM of Y-33075 a reduction in p-MLC was observed and at higher Y-33075 concentrations this reduction of p-MLC became even more apparent. Treatment with Y-27632 similarly decreased phosphorylation of MLC starting at a concentration of 100 nM and thus proving a higher drug concentration of Y-27632 necessary to obtain a similar effect to Y-33075 (Fig 1G and 1H).

To determine whether the effect observed on contraction was associated with the inhibition of the Rho-kinase and subsequently less collagen expression, we examined the protein levels of the phosphorylated ROCK substrate Moesin and collagen 1a1. Therefore, TWNT-4 and FVB/ NJ HSCs were incubated with both ROCK inhibitors for 24 hours before being snap frozen, processed and analyzed. Both ROCK inhibitors, Y-27632 and Y-33075, reduced Moesin phosphorylation and protein expression of collagen 1a1 in human and murine HSCs (Fig 2A–2H). In TWNT-4 cells the collagen 1a1 expression was reduced by Y-33075 at all concentrations, whereas Y-27632 significantly decreased collagen 1a1 expression only at the highest concentration of 10 μM (Fig 2A and 2B). This result was confirmed by the gene expression level of *Col1a1* detected by qRT-PCR in TWNT-4 cells. Again, both Rho-kinase inhibitors (Y-33075 and Y-27632) reduced *Col1a1* gene expression at a concentration of 1 μM while Y-33075 also displayed a significant reduction at a 10 μM concentration (Fig 2I). In primary isolated FVB/ NJ HSCs, the protein expression of the substrate p-Moesin showed a tendency towards reduction starting with the concentration of 1 μM for Y-33075 and a significant reduction at all concentrations of Y-27632 (Fig 2E–2H). Y-33075 reduced *Col1a1* expression across all concentrations in a dose dependent manner and p-Moesin showed a significant reduction at the highest concentration (Fig 2E, 2F–2H). The ROCK inhibitor Y-27632 reduced the col1a1 protein expression starting with a concentration of 100 nM (Fig 2E and 2F). Likewise primary murine FVB/NJ cells demonstrated a significant reduction in Col1a1 mRNA expression after

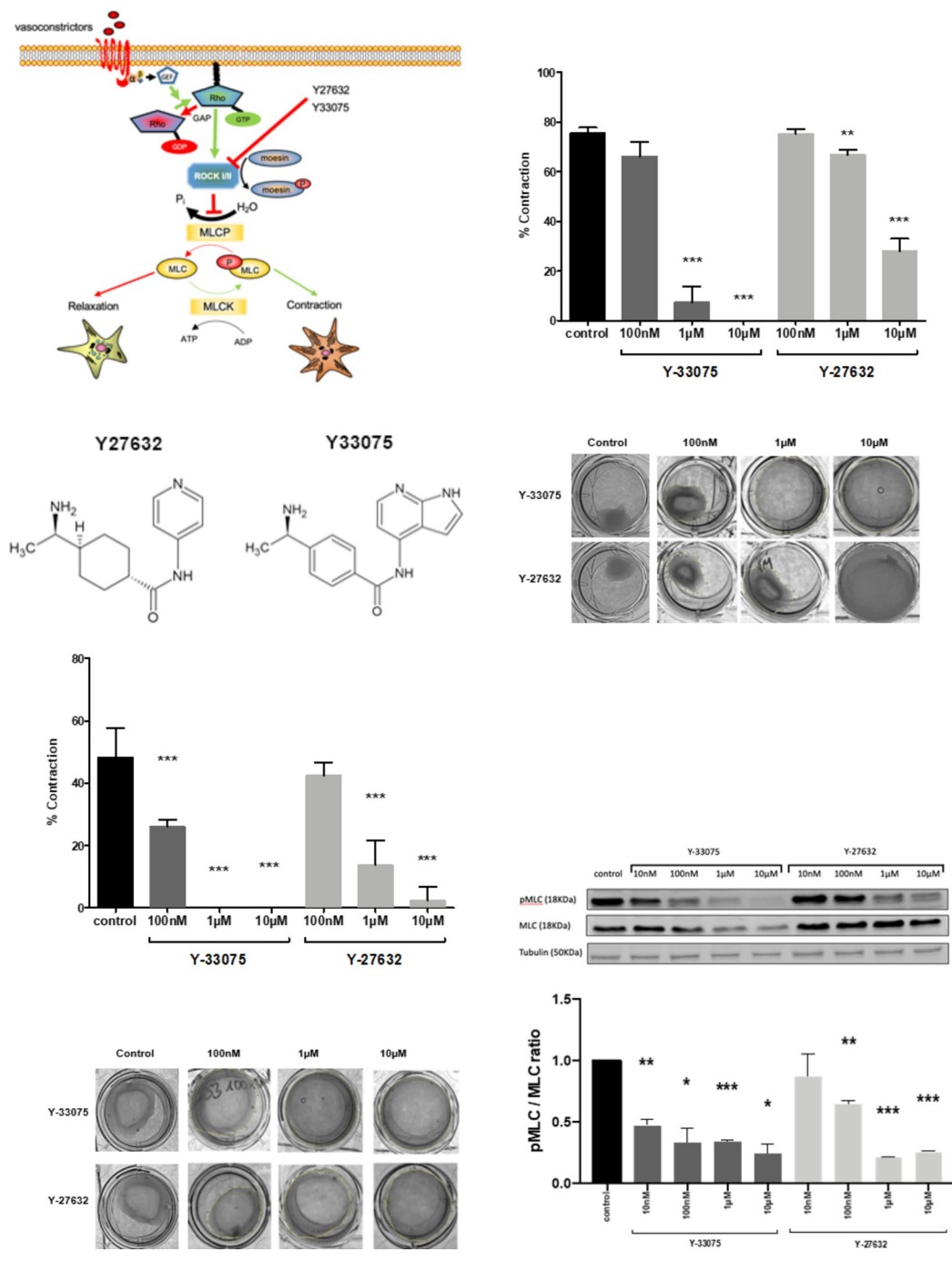

**Fig 1. Y-27632 and Y-33075 treatments reduce contraction in human TWNT-4 and primary murine HSCs.** (A) Schematic depiction of the RhoA-ROCK pathway including mechanism of action of the ROCK inhibitors Y-27632 and Y-33075. (B) Chemical structure of ROCK inhibitors Y-27632 and Y33075. (C) Effect of Y-27632 and Y-33075 on collagen gel contraction of activated human TWNT-4 cells (n = 3). (D) Representative images of the contraction assay in TWNT-4 cells after incubation with the ROCK inhibitors for 24 hours. (E) Effect of Y-27632 and Y-33075 on collagen gel contraction of culture-activated primary murine FVB/NJ cells (n = 3). (F) Representative images of gel contraction in FVB/NJ cells 24 hours after treatment with the ROCK inhibitors. (G) Protein expression levels of phospho-MLC and MLC in TWNT-4 after 24 hour incubation with the ROCK inhibitors. (H) Ratio of phospho-MLC to MLC in TWNT-4 cell lysates after treatment with ROCK inhibitors for 24 hours (n = 2). Results are expressed as the mean ± standard error of the mean (SEM); *p<0.05, **p<0.01 and ***p<0.001.

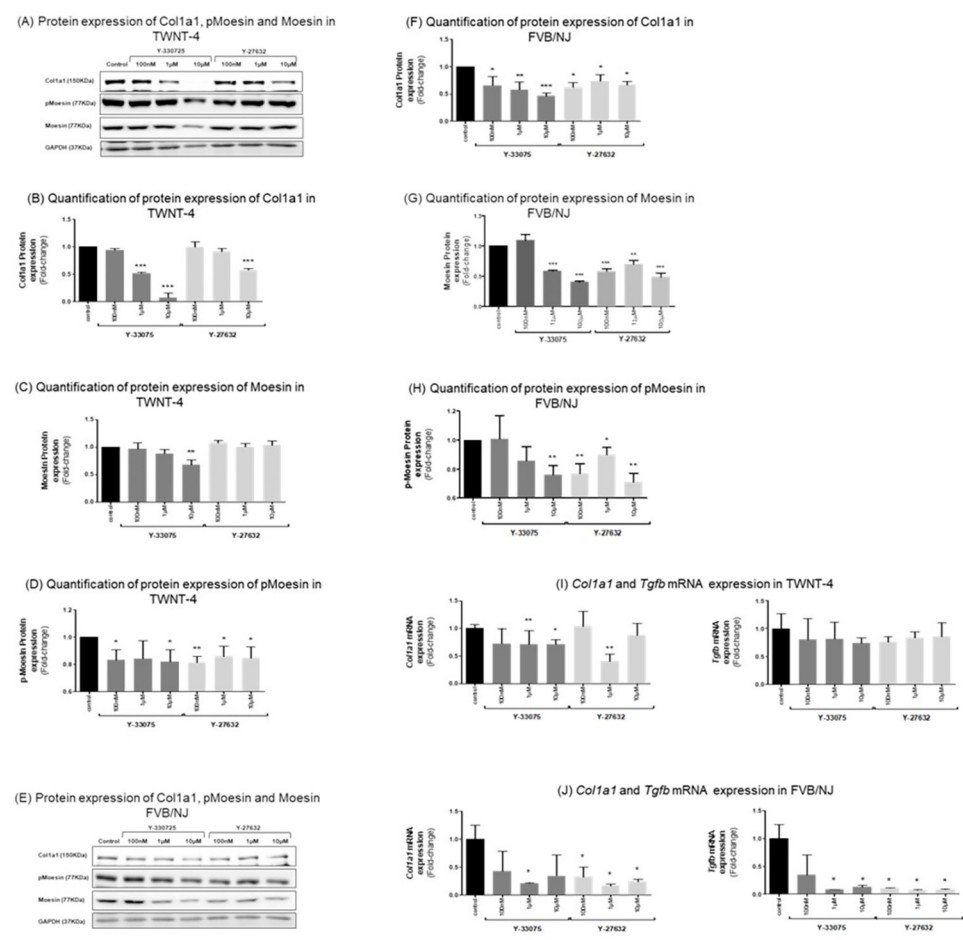

**Fig 2. Y-27632 and Y-33075 treatments reduce fibrosis in human TWNT-4 and primary mouse HSCs.** (A) Western blots of Col1a1, p-Moesin, Moesin and GAPDH using TWNT-4 cell lysates after 24 hour incubation with the ROCK inhibitors. (B) Quantification of protein expression of Col1a1 after treatment of TWNT-4 cells with ROCK inhibitors for 24 hours (n = 3). (C) Quantification of protein expression of Moesin after treatment of TWNT-4 cells with ROCK inhibitors for 24 hours (n = 3). (D) Quantification of protein expression of pMoesin after treatment of TWNT-4 cells with ROCK inhibitors for 24 hours (n = 3). (E) Western Blots of Col1a1, p-Moesin, Moesin and GAPDH using cell lysates of primary FVB/NJ cells after incubation with ROCK inhibitors for 24 hours. (F). Quantification of protein expression of Col1a1 after treatment of FVB/NJ cells with ROCK inhibitors for 24 hours (n = 3). (G) Quantification of protein expression of Moesin after treatment of FVB/NJ cells with ROCK inhibitors for 24 hours (n = 3). (H) Quantification of protein expression of p-Moesin after treatment of FVB/NJ cells with ROCK inhibitors for 24 hours (n = 3). (I) TWNT-4 mRNA expression levels of the pro-fibrotic marker *Col1a1* and *Tgfβ* (corrected to *18s* as a housekeeping gene) after 24h incubation with ROCK inhibitors Y-27632 and Y-33075 (n = 3). (J) FVB/NJ mRNA expression levels of the pro-fibrotic marker *Col1a1* and *Tgfβ* (corrected to *18s* as a housekeeping gene) after 24h incubation with ROCK inhibitors Y-27632 and Y-33075 (n = 3). Results are expressed as the mean ± standard error of the mean (SEM); *p<0.05, **p<0.01 and ***p<0.001.

treatment with 1 μM Y-33075 whereas all concentrations of Y-27632 (100 nM, 1 μM, 10 μM) showed a significant decrease (Fig 2J).

The gene expression of *Tgfβ*, as a marker for HSC activation, showed only a tendency towards reduction after treatment of TWNT-4 cells with Y-33075 or Y-27632 (Fig 2I). However, in primary murine FVB/NJ cells this effect was far more pronounced with concentrations of 1 μM and 10 μM of Y-33075 as well as all concentrations of Y-27632 (100 nM, 1 μM, 10 μM) displaying a drastic reduction in Tgfβ mRNA expression (Fig 2J).

## ROCK inhibitors increase migration in primary mouse HSCs

To evaluate functional effects of Y-33075 and Y-27632 on HSCs, a wound healing assay was performed in the presence or absence of Y-33075 or Y-27632 (100 nM, 1 µM, 10 µM). Remarkably, migration was increased after Y-33075 incubation (1 µM) at 4, 8 and 24h in TWNT-4 cells. On the other hand, the concentration of 10 µM Y-33075 decreased TWNT-4 migration after 24h significantly. The Rho-kinase inhibitor Y-27632 (10 µM) also increased the migration of TWNT-4 cells after 4 and 8h of incubation. Furthermore, the migration of primary mouse FVB/NJ HSCs was increased by 100 nM, 1 µM and 10 µM Y-33075 after 4 and 8h of incubation (Fig 3A and 3B). Incubation of 24h with Y-33075 (1 µM and 10 µM) also increased FVB/NJ HSCs migration significantly. There was no significant effect of Y-27632 on FVB/NJ HSCs migration observed after 4h of incubation. However, 1 µM and 10 µM of Y-27632 significantly

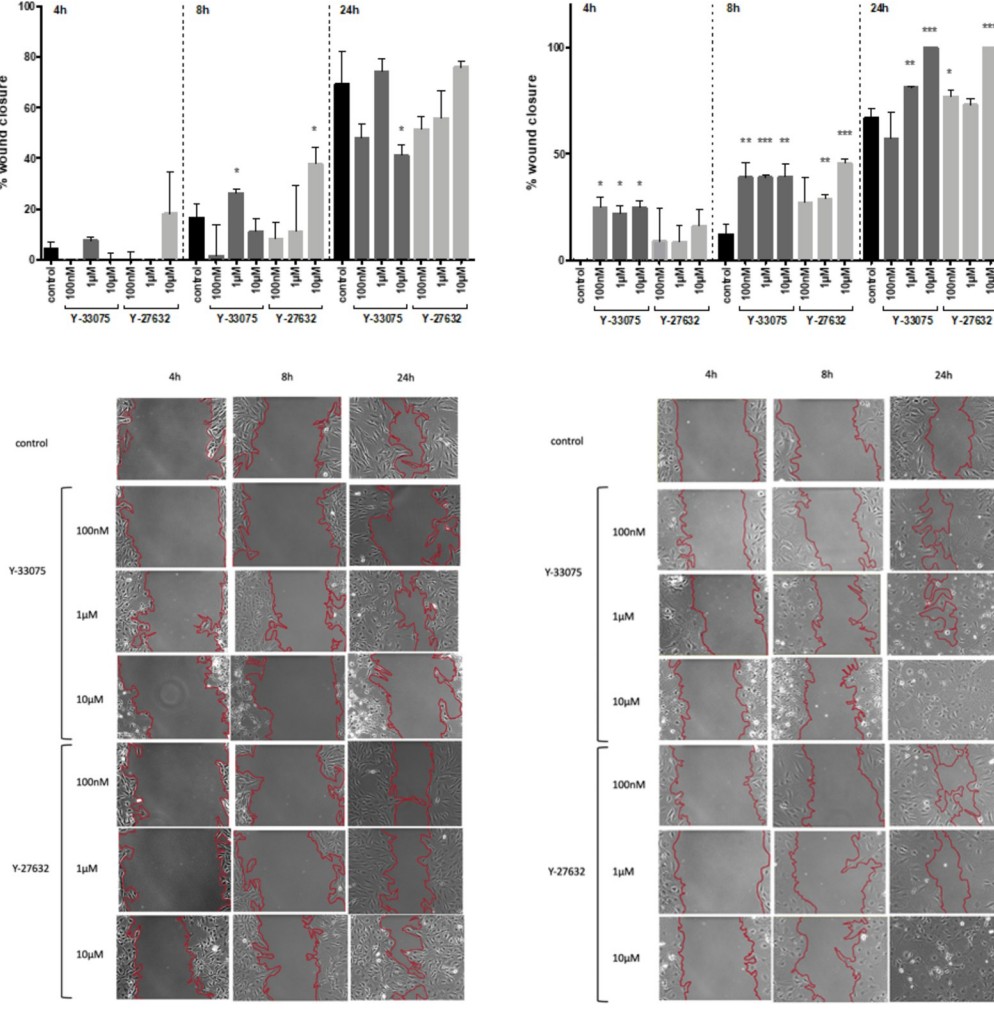

**Fig 3. Y-27632 and Y-33075 treatments increase migration in human TWNT-4 and primary murine HSCs.** (A) Effect of ROCK inhibitors Y-27632 and Y-33075 on wound closure in activated human TWNT-4 cells measured at 4, 8 and 24 hours (n = 3). (B) Representative images of the wound healing assay in TWNT-4 cells at 4, 8 and 24 hours after treatment with the ROCK inhibitors. The scale bars apply for all images and the length represents 200µm. (C) Effect of ROCK inhibitors Y-27632 and Y-33075 on wound closure in primary culture-activated FVB/NJ cells measured at 4, 8 and 24 hours (n = 3). (D) Representative images of the wound healing assay taken 4, 8 and 24 hours after treatment of FVB/NJ cells with ROCK inhibitors (scale bars, 200µm). Results are expressed as the mean ± standard error of the mean (SEM); *p<0.05, **p<0.01 and ***p<0.001.

increased the FVB/NJ HSCs migration after 8h. The Y-27632 concentrations of 100 nM and 10 μM significantly increased FVB/NJ HSCs migration after 24h compared to untreated FVB/NJ HSCs (Fig 3C and 3D).

However, in the scratch assay an increase in gap closure rate as seen in our experiments can not only be caused by an increase in migration, but potentially also by an increase in proliferation. Therefore, a BrdU-Assay was performed to investigate the effect of Y-33075 and Y-27632 on TWNT-4 and FVB/NJ HSCs proliferation as an underlying cause for the increase in wound healing.

## ROCK inhibitors decrease proliferation in both human and murine HSCs

Using a BrdU-Assay, the effect of the ROCK inhibitors on proliferation in human and murine HSCs was investigated. The ROCK inhibitor Y-33075 significantly decreased the proliferation of TWNT-4 HSCs at concentrations of 100 nM, 1 μM and 10 μM (Fig 4A), while Y-27632 significantly decreased the proliferation of TWNT-4 cells at concentration of 1 μM and above. These findings were further confirmed by analyzing *PCNA* mRNA expression levels which only demonstrated the tendency towards decrease at these concentrations (Fig 4B). In murine FVB/NJ HSCs both Rho-kinase inhibitors reduced proliferation at all concentrations (100 nM, 1 μM, 10 μM) as shown by a decreased BrdU content (Fig 4C). These results were further substantiated by the significant reduction of Pcna mRNA expression after treatment of FVB/NJ cells with either of the ROCK inhibitors at concentrations of 100 nM, 1 μM and 10 μM (Fig 4D).

## ROCK inhibitors reduce the activation of HSC

Both, Y-33075 and Y-27632, decreased the protein expression of αSMA, a marker for the activation and transdifferentiation of HSCs to myofibroblastic cells. The Western Blots showed a tendency towards reduction of the αSMA expression in TWNT-4 cells by both Y-33075 and Y-27632 at a concentration of 1 μM and a significant decrease at a 10 μM concentration (Fig 4E and 4F). On the other hand, only 10 μM of the inhibitor Y-33075 decreased αSMA expression in murine FVB/NJ HSCs. Y-27632 displayed a significant decrease across all concentrations in αSMA protein expression was found in FVB/NJ HSCs (Fig 4G and 4H).

## Discussion

The present study examined the effects of the ROCK inhibitors Y-27632 and Y-33075 on fully activated and transdifferentiated myofibroblastic HSCs of both murine and human origin. We could confirm that both inhibitors significantly reduced contraction and fibrogenesis in primary mouse HSC, as well as the human TWNT-4 cell line. These results are in line with previous publications where authors demonstrated that inhibition of the Rho-kinase showed beneficial effects in reducing the formation of fibrosis *in vitro* and portal hypertension *in vivo* [25–27].

Furthermore, our study allowed a direct comparison between both ROCK inhibitors at a range of concentrations, thus giving valuable insight into their efficacy. We confirmed that Y-33075 is approximately 10-fold more potent than Y-27632 which is consistent with previous findings in different cell types [28]. Due to the fact, that ROCK inhibition also plays a role in macrophage function, we focused in this study on *in vitro* experiments with solely HSCs [29]. Therefore, our *in vitro* data highlight the exact role of ROCK inhibition in HSC without other cellular influences, which of course has the limitation of neglecting cell-cell interactions.

Previously it was shown, that after stimulation with PDGF or ET-1, inhibition of Rho-kinase with Y-27632 inhibited migration in primary rat HSCs [26, 30]. By contrast, in our

## (A) BrdU in TWNT-4

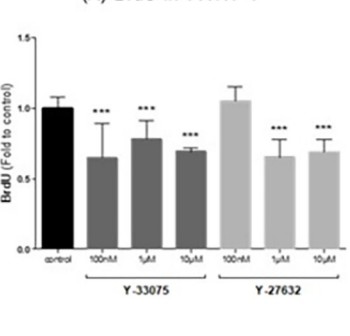

## (E) Protein expression of α-SMA in TWNT-4

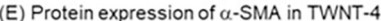
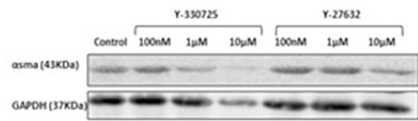

## (B) *Pcna* mRNA expression in TWNT-4

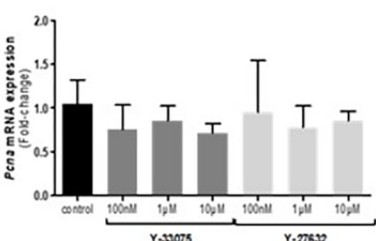

## (F) Quantification of protein expression of α-SMA in TWNT-4

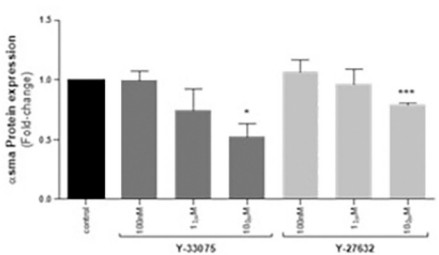

## (C) BrdU in FVB/NJ

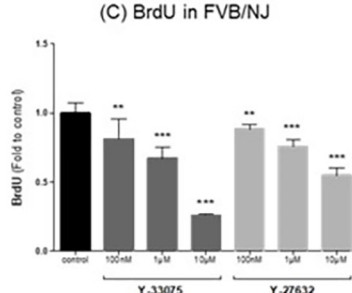

## (G) Protein expression of α-SMA in FVB/NJ

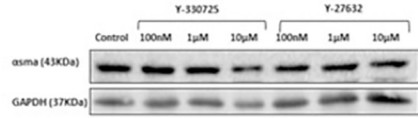

## (D) *Pcna* mRNA expression in FVB/NJ

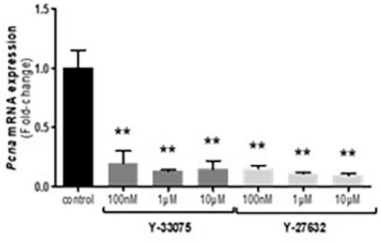

## (H) Quantification of protein expression of α-SMA in FVB/NJ

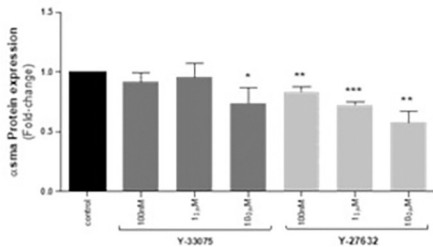

**Fig 4. Y-27632 and Y-33075 treatments decrease proliferation in human TWNT-4 and primary mouse HSCs.** (A) Colorimetric BrdU evaluation of proliferation levels in TWNT-4 cells after 24h treatment with ROCK inhibitors (n = 3). (B) mRNA expression of *Pcna* as a marker for proliferation in TWNT-4 cells (corrected against *18s* as a housekeeping gene) after 24 hour treatment with ROCK inhibitors. (C) Effect of 24 hour ROCK inhibitor treatments on proliferation in murine FVB/NJ cells assessed by colorimetric BrdU-Assay (n = 3). (D) mRNA expression of *Pcna* as a marker for proliferation in FVB/NJ cell lysates (corrected against *18s* as a housekeeping gene) after 24 hour treatment with ROCK inhibitors (n = 3). (E) Protein expression of α-SMA as a marker for the activation of HSCs using cell lysates of human TWNT-4 cells after 24 hour treatment with Y-27632 and Y-33075. (F) Quantification of α-SMA protein expression after treatment of TWNT-4 cells with ROCK inhibitors for 24 hours (n = 3). (G) α-SMA protein

expression in murine FVB/NJ cell lysates as a marker for activation in HSCs after 24 hour treatment with ROCK inhibitors. (H) Quantification of α-SMA protein expression after treatment of TWNT-4 cells with ROCK inhibitors for 24 hours (n = 3). Results are expressed as the mean ± standard error of the mean (SEM); *p<0.05, **p<0.01 and ***p<0.001.

wound-healing assay treatment with the ROCK inhibitors Y-27632, as well as Y-33075 increased migration in FVB/NJ cells after 4h, 8h and 24h. Given these conflicting results, the role of ROCK inhibition in the behavior of HSCs in liver fibrosis appears more diverse than previously assumed. Thus, it would be an important approach to examine migration in *in vivo* activated HSCs. However, we speculate that also in vivo activated HSC would react in the same manner as our investigated cells. The scope of this study is to inhibit both isoforms of ROCK, ROCK1 and ROCK2. While ROCK1 promotes the formation of stress fibres and focal adhesions, ROCK2 is known to inhibit the actin polymerization and thus migration. Therefore the specific inhibition of only ROCK1 or ROCK2 in HSC remains to be elucidated. However, both isoforms contribute to the profibrotic phenotype of HSC, thus we used non specific ROCK inhibitors.

A similar ambivalence towards migration after treatment with ROCK inhibitors has been described in previous studies of other cell types most notably cancer cells. Here extensive research has displayed both, a decrease in motility for example in ovarian, breast and lung cancer cells [31–33], as well as an increase in some cancer types such as skin, colon and breast carcinoma cells [34–37]. However, the pro-migratory effects of ROCK inhibition is not limited to cancer cells but also exhibited in a multitude of non-neoplastic cell types e.g. microglial, retinal pigment epithelial or human periodontal ligament stem cells [38–40].

This myriad of conflicting results further illustrates the exceptionally complex role ROCK plays in the regulation of cell motility.

In light of these ambiguities, a closer look at the role of the isoforms ROCK1 and ROCK2 is warranted. Historically it has been assumed that both isoforms play a similar role in cell functions. However, more recently several important type-dependent discrepancies in their effects on cytoskeleton modulation, cell morphology, adhesion and invadopedia have been described [41]. For instance ROCK1 knock down (KD) cells showed a reduction in focal adhesion and migration, while ROCK2 KD cells did not display this phenotype [42]. Similar results were exhibited in glioblastoma cells where ROCK2 KD increased migration by increasing phosphorylation of Cdc42/Rac which regulates actin polymerization and thus plays a central role in cell migration [43]. It follows that the effects of ROCK inhibition on cell motility could be determined by which ROCK isoform a pan-ROCK inhibitor such as Y-27632 and Y-33075 predominately inhibits and thus be cardinally linked to the current expression pattern of the cell in question.

Alternatively the increase in wound healing after Y-27632 treatment has previously been ascribed to an increase in cell spreading by means of exorbitant cell protrusion rather than an increase in directed migration [44] and this hypothesis likewise deserves additional investigation.

However, this study has its limitations, since we focused on the inhibition of both ROCK isoforms. Thus the role of the specific ROCK isoforms on HSC migration may be investigated in future studies. Y-27632 and Y-33075 also have off-target effects which need to be taken into account. However, it has been shown, that the affinity of those ROCK inhibitors is 200–2000 times to ROCK over other kinases and the concentrations used in this study are too low to induce off-target effects.

In the context of fibrogenesis, both Rho-kinase inhibitors decreased col1a1 and aSMA protein and gene expression in human and murine HSCs. Therefore, both Rho-kinase inhibitors

reduce the fibrogenic phenotype of the investigated HSCs. Moreover, considering the decrease in hepatic resistance demonstrated in cirrhotic bile duct ligated (BDL) rats [45] after ROCK inhibition with Y-27632, it is yet unclear whether the increased motility of HSCs may lead to migration out of the fibrotic septa and thus reduce the formation of fibrosis.

Our *in vitro* data may deliver novel insights of the role of ROCK inhibition in liver fibrosis. We hypothesize that the increase in migration of non-fibrogenic HSCs is potentially beneficial in liver fibrosis and as such that ROCK inhibition is a suitable approach for treatment.

## Supporting information

**S1 File.**
(PDF)

**S1 Table. Antibodies.**
(DOCX)

## Acknowledgments

The authors thank Evelyn Süß and Dikra Zouiten for excellent technical assistance.

## Author Contributions

**Conceptualization:** Nadine Bachtler, Sandra Torres, Cristina Ortiz, Robert Schierwagen, Olaf Tyc, Christoph Hieber, Marie-Luise Berres, Caroline Meier, Nico Kraus, Stefan Zeuzem, Bart Nijmeijer, Sebas Pronk, Jonel Trebicka, Sabine Klein.

**Data curation:** Nadine Bachtler, Sandra Torres, Cristina Ortiz, Robert Schierwagen, Olaf Tyc, Christoph Hieber, Marie-Luise Berres, Caroline Meier, Nico Kraus, Stefan Zeuzem, Sebas Pronk, Sabine Klein.

**Formal analysis:** Nadine Bachtler, Sandra Torres, Cristina Ortiz, Robert Schierwagen, Olaf Tyc, Christoph Hieber, Marie-Luise Berres, Caroline Meier, Nico Kraus, Sebas Pronk, Sabine Klein.

**Funding acquisition:** Jonel Trebicka.

**Investigation:** Jonel Trebicka, Sabine Klein.

**Methodology:** Nadine Bachtler, Sandra Torres, Cristina Ortiz, Robert Schierwagen, Olaf Tyc, Marie-Luise Berres, Caroline Meier, Nico Kraus, Sebas Pronk, Sabine Klein.

**Supervision:** Stefan Zeuzem, Bart Nijmeijer, Jonel Trebicka, Sabine Klein.

**Validation:** Stefan Zeuzem, Bart Nijmeijer, Jonel Trebicka, Sabine Klein.

**Writing – original draft:** Nadine Bachtler, Sandra Torres, Bart Nijmeijer, Jonel Trebicka, Sabine Klein.

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
