## [Decision Letter · Decision Letter 0]

8 Mar 2022

PONE-D-22-01789Rho-kinase inhibitors Y-27632 and Y-33075 decrease contraction but increase migration in murine and human hepatic stellate cells.PLOS ONE

Dear Dr. Trebicka,

Thank you for submitting your manuscript to PLOS ONE. After careful consideration, we feel that it has merit but does not fully meet PLOS ONE’s publication criteria as it currently stands. Therefore, we invite you to submit a revised version of the manuscript that addresses the points raised during the review process.

We look forward to receiving your revised manuscript.

Kind regards,

Michael F Olson, PhD

Academic Editor

PLOS ONE

Journal Requirements:

"This study was supported by the German Research Foundation (DFG) project ID 403224013 – SFB 1382 (A09), by the German Federal Ministry of Education and Research (BMBF) for the DEEP-HCC project and by the Hessian Ministry of Higher Education, Research and the Arts (HMWK) for the ENABLE cluster project and by Eurostars (Grant ID 12350). The MICROB-PREDICT (project ID 825694), DECISION (project ID 847949), GALAXY (project ID 668031), LIVERHOPE (project ID 731875) and IHMCSA (project ID 964590) projects have received funding from the European Union’s Horizon 2020 research and innovation program. The manuscript reflects only the authors’ views, and the European Commission is not responsible for any use that may be made of the information it contains. The funders had no influence on study design, data collection and analysis, decision to publish, or preparation of the manuscript."

We note that you have provided funding information. However, funding information should not appear in the Funding section or other areas of your manuscript. We will only publish funding information present in the Funding Statement section of the online submission form. 

"This study was supported by the German Research Foundation (DFG) project ID 403224013 – SFB 1382 (A09), by the German Federal Ministry of Education and Research (BMBF) for the DEEP-HCC project and by the Hessian Ministry of Higher Education, Research and the Arts (HMWK) for the ENABLE cluster project and by Eurostars (Grant ID 12350). The MICROB-PREDICT (project ID 825694), DECISION (project ID 847949), GALAXY (project ID 668031), LIVERHOPE (project ID 731875) and IHMCSA (project ID 964590) projects have received funding from the European Union’s Horizon 2020 research and innovation program. The manuscript reflects only the authors’ views, and the European Commission is not responsible for any use that may be made of the information it contains. The funders had no influence on study design, data collection and analysis, decision to publish, or preparation of the manuscript."

Additional Editor Comments:

The manuscript has been reviewed by two experts in the field, both of whom have some suggested changes/modifications. In particular, please note some of their questions that should be addressed in the discussion, including the potential roles of ROCK1 vs ROCK2, and the in vivo relevance of the observations.

Reviewers' comments:

Reviewer's Responses to Questions

**Comments to the Author**

1. Is the manuscript technically sound, and do the data support the conclusions?

Reviewer #1: Yes

Reviewer #2: Yes

2. Has the statistical analysis been performed appropriately and rigorously? 

Reviewer #1: Yes

Reviewer #2: Yes

3. Have the authors made all data underlying the findings in their manuscript fully available?

Reviewer #1: Yes

Reviewer #2: Yes

4. Is the manuscript presented in an intelligible fashion and written in standard English?

Reviewer #1: Yes

Reviewer #2: Yes

5. Review Comments to the Author

Reviewer #1: This study investigated the effects of ROCK chemical inhibitors, including Y-33075 and Y-27632, on contraction, proliferation, and migration in primary isolated hepatic stellate cell (HSC) from FVB mice and human HSC line TWNT-4. The author found that bot ROCK inhibitors suppressed contraction, proliferation, and fibrogenesis in primary isolated HSC and TWNT-4 cells. However, increased migration of HSC was found when both inhibitors were treated. They also pointed out that Y-33075 shows a 10-times potency compared to Y-27632. Some of data are consistent with the previous findings but some of it produce conflicting results, at the author indicated.

The inhibitors the authors used are not specific for ROCK1 or ROCK2. The question is which isoform could play a key role in regulating contraction, proliferation, and migration in the current settings. At least, the authors need to study selective specific ROCK2 inhibitor in the current experimental settings. It has reported that ROCK2 chemical inhibitor is quite specific.

The limitation of the current study is a lack of in vivo data. The authors may discuss this limitation in the section of discussion.

Reviewer #2: The manuscript by Bachtler and colleagues examined the effects of ROCK inhibition on contraction, migration and fibrogenic markers in murine and human hepatic stellate cells. The authors suggest that ROCK inhibition should be evaluated as a treatment strategy for hepatic fibrosis.

Various studies examining the effects of ROCK inhibition on cell contraction and migration have demonstrated some contrasting results and this manuscript aimed to compare two ROCK inhibitors in both human and murine HSC. The manuscript would have been strengthened by examination of the ROCK isoforms independently. HSC isolated from mice with hepatic fibrosis rather than culture-activated HSC or in vivo studies may have also provided further insights into the mechanisms and clinical utility of ROCK inhibition.

1) Are the ROCK inhibitors studied highly specific? Do they result in any off-target effects that may influence the results described? Have alternative methods of ROCK inhibition been performed to confirm the results shown with the inhibitors? Inhibition of ROCK isoforms would have strengthened the manuscript.

2) Can the authors provide an explanation for the apparent differences observed between murine and human cells regarding Tgfb and SMA expression following ROCK inhibition?

3) Quantification of western blots was performed in Figure 1H but has not been provided for Figures 2A, B and 4E and F. It is a little difficult to observe the described decrease in p-moesin following Y-27632 in TWNT-4 cells. Was quantification performed and were these changes described for Figure 2 and 4 significant?

4) Scale bars are required on images in Figures 3B and D.

5) The number of biological replicates performed for each experiment should be indicated in the figure legends.

6) In Figure 4B the Pcna gene expression in TWNT-4 cells is not significantly altered by ROCK inhibitors therefore the statement made by the authors that this result confirms the BrdU results in Fig4A is not entirely true. Some rewording of this sentence is required to avoid over-interpretation of the results.

7) Previous reports examining ROCK inhibition following stimulation of HSC and this study using only culture-activated HSC demonstrate different results in regards to HSC migration. Can the authors speculate on how ROCK inhibition in HSC isolated from fibrotic livers would have impacted migration? Might this be a worthwhile approach?

6. PLOS authors have the option to publish the peer review history of their article (what does this mean?). If published, this will include your full peer review and any attached files.

Reviewer #1: No

Reviewer #2: No

---

## [Author Response · Author response to Decision Letter 0]

19 Apr 2022

Response to Reviewers

Reviewer #1: This study investigated the effects of ROCK chemical inhibitors, including Y-33075 and Y-27632, on contraction, proliferation, and migration in primary isolated hepatic stellate cell (HSC) from FVB mice and human HSC line TWNT-4. The author found that bot ROCK inhibitors suppressed contraction, proliferation, and fibrogenesis in primary isolated HSC and TWNT-4 cells. However, increased migration of HSC was found when both inhibitors were treated. They also pointed out that Y-33075 shows a 10-times potency compared to Y-27632. Some of data are consistent with the previous findings but some of it produce conflicting results, at the author indicated.

The inhibitors the authors used are not specific for ROCK1 or ROCK2. The question is which isoform could play a key role in regulating contraction, proliferation, and migration in the current settings. At least, the authors need to study selective specific ROCK2 inhibitor in the current experimental settings. It has reported that ROCK2 chemical inhibitor is quite specific.

We thank the reviewer for this comment. We fully agree that the inhibitors used in this study affect both isoforms, ROCK I and ROCK II, which was also the goal of our study. We apologize for not making this clear enough in the manuscript. Indeed ROCK1 and 2 have very different functions and both contribute to the profibrotic phenotype of HSC.

ROCK I initiates cell polarity by the formation of actomyosin filament bundles, promoting the formation of stress fibres and focal adhesions, while ROCK II controls migration by inhibiting actin polymerization (Niwell.-Litwa K. et al., ROCK 1 and 2 differentially regulate actomyosin organization to drive cell and synaptic polarity. J. Cell Biol. 210 (2015) 225-242). Regarding fibrosis, it could be published that the deletion of ROCK II in cardiac fibroblasts and thus reducing hypertrophy and fibroses. ROCK II, but not ROCK I regulates TGF-beta induced fibrotic gene expression. Therefore, these complementary effects of ROCK1 and 2 require a non-specific inhibitor, as performed in our experimental setup. We have clarified this in the manuscript (see page 12, lines 280 – 285) and changed the title accordingly (see page 1, line 1). We also included a statement that the selective inhibition of the ROCK isoforms is beyond the scope of the manuscript and that this should be studied in the future. See page 13, lines 312-314.

The limitation of the current study is a lack of in vivo data. The authors may discuss this limitation in the section of discussion.

Thank you for this comment. 

In this study we excluded in vivo data, since the effects of the ROCK inhibitor Y-27632 have already been extensively examined in animal models by our group and others (Hennenberg et al., Defective RhoA/Rho-Kinase Signaling Contributes to Vascular Hypocontractility and Vasodilation in Cirrhotic Rats. GASTROENTEROLOGY 2006;130:838 –854.; Klein et al., HSC-specific inhibition of Rho-kinase reduces portal pressure in cirrhotic rats without major systemic effects., Journal of Hepatology 2012 vol. 57 j 1220–1227.; Klein et al., Rho kinase inhibitor coupled to peptide modified albumin carrier reduces PP and increases perfusion in cirrhotic rats., Rho-kinase inhibitor coupled to peptide-modifed albumin carrier reduces portal pressure and increases renal perfusion in cirrhotic rats., Scientific Reports (2019) 9:2256.; Tada S, Iwamoto H, Nakamuta M, et al. A selective ROCK inhibitor, Y27632, prevents dimethylnitrosamine-induced hepatic fibrosis in rats. J Hepatol. 2001;34(4):529-536.).

These studies demonstrate a significant reduction in portal pressure due to a decrease in intrahepatic vascular resistance, as well as a reduction in fibrosis after treatment of rats with Y-27632. However, recent data have demonstrated that ROCK2 inhibition plays a role in macrophage function and additionally reduces fibrosis (Nalkurthi C et al., ROCK2 inhibition attenuates profibrogenic immune cell function to reverse thioacetamide-induced liver fibrosis., JHEP Reports 2022 vol. 4 j 100386). Therefore, the effect of ROCK inhibitors is not limited to one cell type in the liver. In this manuscript we highlight the exact role of ROCK inhibition only in HSC.

However, we acknowledge that in vitro data, have the limitation of neglecting cell-cell interactions, but was not the goal of our study. We have now included this limitation in the discussion (see page 12, line 271-274 in the “revised manuscript with track changes”).

Reviewer #2: The manuscript by Bachtler and colleagues examined the effects of ROCK inhibition on contraction, migration and fibrogenic markers in murine and human hepatic stellate cells. The authors suggest that ROCK inhibition should be evaluated as a treatment strategy for hepatic fibrosis.

Various studies examining the effects of ROCK inhibition on cell contraction and migration have demonstrated some contrasting results and this manuscript aimed to compare two ROCK inhibitors in both human and murine HSC. The manuscript would have been strengthened by examination of the ROCK isoforms independently. HSC isolated from mice with hepatic fibrosis rather than culture-activated HSC or in vivo studies may have also provided further insights into the mechanisms and clinical utility of ROCK inhibition.

1) Are the ROCK inhibitors studied highly specific? Do they result in any off-target effects that may influence the results described? Have alternative methods of ROCK inhibition been performed to confirm the results shown with the inhibitors? Inhibition of ROCK isoforms would have strengthened the manuscript.

Thank you very much for this comment. The ROCK inhibitors which are studied in this manuscript do not have off-target effects with the concentrations used in this manuscript. 

However, the ROCK inhibitor Y-27632 and also Y-33075 (also known as Y39983) selectively inhibits p160 ROCK and ROCK2, but in very high concentrations it also inhibits other protein kinases like protein kinase C (PKC), cAMP-dependent protein kinase and myosin light-chain kinase. Y-27632 demonstrates a Ki of 0.14µM for p160ROCK whereas Ki values of PKC and cAMP-dependent protein kinase are 26µM and 25 µM respectively. For MLCK the Ki value is as high as 250µM. Y-33075 shows even a stronger inhibition of p160ROCK with the Ki value below 0.005µM. The Ki value for MLCK is 225µM of Y-33075 (Tokushige H. et al., Effects of topical administration of Y-39983, a selective Rho-associated protein kinase inhibitor, on ocular tissues in rabbits and monkeys., Investigative Ophthalmology & Visual Science July 2007, Vol.48, 3216-3222). 

Consequently, Y-27632 and Y-33075 show a �200 times to over 2000 times affinity to ROCK over other kinases. Hence we would consider off target effects minimal. Furthermore the directly conflicting results in migration compared to van Beuge et al [26] cannot be explained by off target effects since the same inhibitor (Y-27632) was used for both experiments (Uehata M et al., Calcium sensitization of smooth muscle mediated by Rho-associated protein kinase in hypertension., Nature. 1997 Oct 30;389(6654):990-4).

These concerns have now been addressed in the discussion of the manuscript (see page 13, lines 314-318). 

We fully agree that the inhibitors used in this study affect both isoforms, ROCK I and ROCK II, which was also the goal of our study. We apologize for not making this clear enough in the manuscript. Indeed ROCK1 and 2 have very different functions and both contribute to the profibrotic phenotype of HSC.

ROCK I initiates cell polarity by the formation of actomyosin filament bundles, promoting the formation of stress fibres and focal adhesions, while ROCK II controls migration by inhibiting actin polymerization (Niwell.-Litwa K. et al., ROCK 1 and 2 differentially regulate actomyosin organization to drive cell and synaptic polarity. J. Cell Biol. 210 (2015) 225-242). Regarding fibrosis, it could be published that the deletion of ROCK II in cardiac fibroblasts and thus reducing hypertrophy and fibroses. ROCK II, but not ROCK I regulates TGF-beta induced fibrotic gene expression. Therefore, these complementary effects of ROCK1 and 2 require a non-specific inhibitor, as performed in our experimental setup. We have clarified this in the manuscript (see page 12, lines 280 – 285) and changed the title accordingly (see page 1, line 1). We also included a statement that the selective inhibition of the ROCK isoforms is beyond the scope of the manuscript and that this should be studied in the future. These concerns have now been addressed in the discussion of the manuscript (see page 13, lines 312-314). 

2) Can the authors provide an explanation for the apparent differences observed between murine and human cells regarding Tgfb and SMA expression following ROCK inhibition?

Thank you for this comment. We would attribute differences in protein expression following ROCK inhibition to the murine cells being primary cells and the human TWNT-4 being a cell line. While cell lines possess key features of primary HSCs they are known to display discrepancies in both their cellular morphology and phenotype, as well as their gene expression and response to growth factors. While primary cells change their phenotype from quiescent and retinoid-storing cells into ECM producing and contractile myofibroblasts, immortalized HSC remain in the phenotypic state. Therefore, the results of the TWNT-4 cells used in this study show the same trend, but not statistical significances as the primary cells (Herrmann J, Gressner AM, Weiskirchen R. Immortal hepatic stellate cell lines: useful tools to study hepatic stellate cell biology and function?. J Cell Mol Med. 2007;11(4):704-722).

3) Quantification of western blots was performed in Figure 1H but has not been provided for Figures 2A, B and 4E and F. It is a little difficult to observe the described decrease in p-moesin following Y-27632 in TWNT-4 cells. Was quantification performed and were these changes described for Figure 2 and 4 significant?

Thank you for the comment and we apologize for the lack of the quantifications of the western blots. We have added the missing quantifications for the Western Blots (figure 2 B, D and figure 4 F, H) to make the results more evident. Moreover, we further specified exact changes and significances in the results section of the manuscript. 

4) Scale bars are required on images in Figures 3B and D.

We apologize for this mistake. The scale bar is now included in the Figures 3B and D. The scale bar represents 200µm, which is described in the revised figure legends. We thank the reviewer for this advice.

5) The number of biological replicates performed for each experiment should be indicated in the figure legends.

Thank you for this comment. Of course this needs to be added in the figure legends to underline the significance. In the revised version we have added the number of biological replicates to the legends for transparency.

6) In Figure 4B the Pcna gene expression in TWNT-4 cells is not significantly altered by ROCK inhibitors therefore the statement made by the authors that this result confirms the BrdU results in Fig4A is not entirely true. Some rewording of this sentence is required to avoid over-interpretation of the results.

We apologize that this was not clear in the original manuscript. We have adjusted the wording to avoid misunderstandings (see page 11, lines 246-247).

7) Previous reports examining ROCK inhibition following stimulation of HSC and this study using only culture-activated HSC demonstrate different results in regards to HSC migration. Can the authors speculate on how ROCK inhibition in HSC isolated from fibrotic livers would have impacted migration? Might this be a worthwhile approach?

This is a very good point. Thank you highlighting this limitation of our study. 

Disparities in gene expression between culture activated and in vivo activated HSCs have previously been described (De Minicis S, Seki E, Uchinami H, et al. Gene expression profiles during hepatic stellate cell activation in culture and in vivo. Gastroenterology. 2007;132(5):1937-1946.).

Thus, it would definitely be important to examine migration in in vivo activated HSCs. However, we believe that the key features of HSC behaviour which were similar in primary HSC as well as in human HSC cell lines would be very similar in in vivo activated HSC. We have highlighted this speculation and limitation in our manuscript (see page 12, lines 281-283).

---

## [Decision Letter · Decision Letter 1]

8 Jun 2022

The non-selective Rho-kinase inhibitors Y-27632 and Y-33075 decrease contraction but increase migration in murine and human hepatic stellate cells.

PONE-D-22-01789R1

Dear Dr. Trebicka,

We apologize for the length of time that has passed before sending you this decision letter, one of the original reviewers agreed to review the revised manuscript but did not perform this task. As a result, an editorial decision has been taken without their input.

We’re pleased to inform you that your manuscript has been judged scientifically suitable for publication and will be formally accepted for publication once it meets all outstanding technical requirements.

Kind regards,

Michael F Olson, PhD

Academic Editor

PLOS ONE

Additional Editor Comments (optional):

Reviewers' comments:

Reviewer's Responses to Questions

**Comments to the Author**

1. If the authors have adequately addressed your comments raised in a previous round of review and you feel that this manuscript is now acceptable for publication, you may indicate that here to bypass the “Comments to the Author” section, enter your conflict of interest statement in the “Confidential to Editor” section, and submit your "Accept" recommendation.

Reviewer #2: (No Response)

2. Is the manuscript technically sound, and do the data support the conclusions?

Reviewer #2: Yes

3. Has the statistical analysis been performed appropriately and rigorously? 

Reviewer #2: Yes

4. Have the authors made all data underlying the findings in their manuscript fully available?

Reviewer #2: Yes

5. Is the manuscript presented in an intelligible fashion and written in standard English?

Reviewer #2: Yes

6. Review Comments to the Author

Reviewer #2: (No Response)

7. PLOS authors have the option to publish the peer review history of their article (what does this mean?). If published, this will include your full peer review and any attached files.

Reviewer #2: No

---

## [Editor Report · Acceptance letter]

10 Jun 2022

PONE-D-22-01789R1 

The non-selective Rho-kinase inhibitors Y-27632 and Y-33075 decrease contraction but increase migration in murine and human hepatic stellate cells. 

Dear Dr. Trebicka:

I'm pleased to inform you that your manuscript has been deemed suitable for publication in PLOS ONE. Congratulations! Your manuscript is now with our production department. 

Kind regards, 

on behalf of

Prof. Michael F Olson 

Academic Editor

PLOS ONE